# Older Adults’ Perspectives of Smart Technologies to Support Aging at Home: Insights from Five World Café Forums

**DOI:** 10.3390/ijerph19137817

**Published:** 2022-06-25

**Authors:** Jackie Street, Helen Barrie, Jaklin Eliott, Lucy Carolan, Fidelma McCorry, Andreas Cebulla, Lyn Phillipson, Kathleen Prokopovich, Scott Hanson-Easey, Teresa Burgess

**Affiliations:** 1School of Public Health, Faculty of Health and Medical Sciences, University of Adelaide, Level 4, Rundle Mall Plaza, 50 Rundle Mall, Adelaide 5000, Australia; jaklin.eliott@adelaide.edu.au (J.E.); scott.hanson-easey@adelaide.edu.au (S.H.-E.); teresa.burgess@adelaide.edu.au (T.B.); 2Australian Centre for Engagement, Evidence and Values, University of Wollongong, Northfields Ave, Wollongong 2522, Australia; lengland@uow.edu.au (L.C.); kar715@uowmail.edu.au (K.P.); 3Centre for Markets, Values and Inclusion, UniSA City West Campus, University of South Australia, Way Lee Building, Adelaide 2072, Australia; helen.barrie@unisa.edu.au; 4Centre of Research Excellence in Translating Nutritional Science to Good Health, University of Adelaide, Level 5, Adelaide Health & Medical Sciences Building, Adelaide 5005, Australia; fidelma.mccorry@adelaide.edu.au; 5Australian Industrial Transformation Institute, College of Business, Government and Law, Flinders University, Adelaide 5001, Australia; andreas.cebulla@flinders.edu.au; 6Faculty of the Arts, Social Science and Humanities, University of Wollongong, Wollongong 2522, Australia; lphillip@uow.edu.au

**Keywords:** artificial intelligence, digital health, smart technologies, gerontechnology, aging in place, older adults, smart home, autonomous vehicles, robotic technologies, smart wearables

## Abstract

Globally, there is an urgent need for solutions that can support our aging populations to live well and reduce the associated economic, social and health burdens. Implementing smart technologies within homes and communities may assist people to live well and ‘age in place’. To date, there has been little consultation with older Australians addressing either the perceived benefits, or the potential social and ethical challenges associated with smart technology use. To address this, we conducted five World Cafés in two Australian states, aiming to capture citizen knowledge about the possibilities and challenges of smart technologies. The participants (*n* = 84) were aged 55 years and over, English-speaking, and living independently. Grounding our analysis in values-based social science and biomedical ethical principles, we identified the themes reflecting the participants’ understanding, resistance, and acceptance of smart technologies, and the ethical principles, including beneficence, non-maleficence, autonomy, privacy, confidentiality, and justice. Similar to other studies, many of the participants demonstrated cautious and conditional acceptance of smart technologies, while identifying concerns about social isolation, breaches of privacy and confidentiality, surveillance, and stigmatization. Attention to understanding and incorporating the values of older citizens will be important for the acceptance and effectiveness of smart technologies for supporting independent and full lives for older citizens.

## 1. Introduction

The Internet of Things (IOT), consisting of smart, interconnected technologies, including smart homes and wearables, is increasingly advocated as a solution for many issues inherent in aging societies [1]. The population demographic aged 65 and over is the fastest growing in Australia, increasing from 12.4% to 16.3% of the population from 2000–2020, and is projected to rise sharply as the baby boomers age [2]. It is estimated that by 2047, 20% of this demographic will be aged 85 plus, an increase from 13% in 2017 [3]. The largest increase, however, is expected in the years to 2037, leaving little time to adjust the support service supply. Already, demographic changes have resulted in increased demand for aged care support with an associated rise in budgetary costs and staffing needs. Given that residential aged care is expensive [4], keeping people at home as long as possible through the use of community care may be both cheaper and more desirable to many. Therefore, Australian policy-makers are considering how best to support individuals to remain in the community as they age [5]. Smart technologies may offer low cost, sustainable solutions for maintaining support and safety for this group, and have been adopted [6,7] or trialed [8] in some community aged-care settings.

Most smart technologies fall into four categories: autonomous vehicles (AVs); smart wearables; smart homes; and robotics; all of them rely on artificial intelligence (AI) and are potentially part of an Internet of Things (IOT): that is, they connect to the internet and are capable of sharing data with each other and with care providers. Smart technologies use AI to predict or detect events: for example, a smart car can detect an object and stop the car before a collision occurs. Similarly, a smart home could be programmed for an individual’s expected timetable of activities so that failure to use the kettle or refrigerator in a timeframe may alert a carer that the person is at risk. Machine learning can take this even further: AI-powered detectors can report falls, but machine learning analytics and wearable sensors may one day be able to personalize care to predict, report, and prevent an imminent risk of falling.

To date, the design and implementation of smart technologies for the proposed benefit of older people has occurred with little consultation with either the organizations working in the sector, or with older people themselves [9,10]. This may mean that the social and ethical challenges that these technologies create for an older population are neither given appropriate attention nor carefully integrated with the values and needs of the users. Previous research suggests that technologies may be abandoned if consultation with the user and matching of the technologies with users’ needs are inadequate [11,12].

There are several challenges to the widespread adoption of smart technologies in the Australian population aged 65 plus. Among the 3.7 million older Australians in 2018 living at home, 38.4% had not used the internet in the previous three months, with higher numbers in the age groups currently most likely to use smart technologies for aging in place (51.5% of those aged 75–84 and 73.3% of those aged 85+) [13]. This was despite 93% of older Australians in May 2021 having access to “internet in their home” [14]. The gap between internet access and actual use likely reflects the view that older people find new technologies overwhelming [8] and “changing so fast it is difficult to keep up with it” (p. 9) [14]. This raises concerns about a digital divide and what Manor and Herscovici term ‘digital ageism’ [9].

The successful roll-out of smart technologies into community and aged care settings will be complex and challenging. It will benefit from input by technical experts and the perspectives of the potential future users, who can best identify the benefits and drawbacks of the particular technological solutions offered. In this research, we aimed to capture, through guided conversations with older citizens, the nuanced deliberations of a diverse group regarding the principal social and ethical challenges posed, creating new knowledge about the possibilities and challenges of using smart technologies to support older people to live comfortably and well in their own homes.

The World Café approach, used in this study, was developed in the 1990s by Juanita Brown and David Isaacs, in collaboration with others [15]. This development paralleled the deliberative turn in democratic theory, described by Dryzek [16] who argued that democratic legitimacy is supported by authentic deliberations on the part of those affected by a collective decision. In using the World Café method, we brought together notions of an informed public, rich collaborative discussion in a hospitable space, and collective knowledge made visible. In this setting, we asked a diverse group of older people to reflect on the actual and potential benefits and harms of using smart technologies in their lives as they age.

## 2. Materials and Methods

### 2.1. Ethics Approval

Ethics approval for the project was provided by the Human Research Ethics Committees of the University of Wollongong (UOW), University of Adelaide, (UA), and the South Australian (SA) Department of Health and Wellbeing.

### 2.2. Participant Recruitment

The participants were aged 55 years and over, living independently, and English-speaking. The targeted recruitment was through relevant newsletters, radio interviews, and organizations including social and sports clubs, churches, public libraries, and aged care providers. In both the recruitment approaches and selection of sites, we aimed for diversity but also purposely recruited participants from low socioeconomic backgrounds, since these individuals, in general, have less input into public policy and fewer choices than individuals from high socioeconomic groups. The participants received an information sheet and consent form, were asked to inform eligible friends and family about the study, and received a $40 honorarium in recognition of their time contribution.

### 2.3. Approach

This study was grounded in values-based social science, or empirical ethics, using deliberative democracy approaches and thematic analysis to understand the perspectives of older people living in the community. The multidisciplinary research team was drawn from UOW and UA and covered a breadth of disciplinary and content expertise, including ethics, community engagement, social economics, medical sociology, gerontology, ergonomics, occupational health, social innovation, design anthropology, chronic disease, and dementia. We used ethical theory to interpret and contextualize these empirical findings and draw normative conclusions about the emerging area of the use of smart technologies in community-based care for the aging [17,18].

### 2.4. Data Collection

Three World Cafés (henceforth Café) were conducted in SA mid-2019, conducted by the UA research team led by one author (HB), with another from UOW (JS) attending two of the Cafés. In NSW, two Cafés were held in November 2019, conducted by the UOW research team led by one author (JS) and hosted by a second author (LC) (see Table 1).

### 2.5. World Café Methodology

In support of authentic deliberations, Brown and Isaacs [15] indicated that the World Café design should include the following elements:

#### 2.5.1. Exploration of a Question Which Matters

Smart technologies to support aging in place are already being introduced in communities in Australia and overseas, often with little consultation with the end-users. The South Australian Office for the Ageing initiated the study to highlight community views and to develop a framework to support the implementation of smart technologies for older people living in community. The technologies were selected based on an international literature review focused on the perceived benefits and challenges of using smart technology in the context of aging in place [5]. In the Cafés, four categories were discussed: smart homes; smart wearables; robotics (in the home); and autonomous vehicles (AVs).

#### 2.5.2. Creation of a Hospitable Space

The four-hour sessions, in a relaxed café-style venue, included breaks for morning tea and lunch. Small group (*n* = 4–6) discussions of 15–20 min were facilitated by an experienced qualitative researcher around an assigned technology category. The researchers’ roles were to guide the discussion with targeted questions (Box 1) and take notes on the paper tablecloths, encouraging participants to do the same. The participants cycled through the tables to cover all of the categories.

Box 1Targeted questions used to frame the discussions
What are some of the good things that could happen from having this technology?What are some of the things you would NOT like about this technology?Would smart technologies make life better or more complicated?Think about what smart technologies might cost you—perhaps in terms of money, privacy, in having to learn how to use them, in changing the way you live your life…Would smart technologies be worth it, if their use meant you could stay at home longer?


#### 2.5.3. Background Information for Participants

We expected that many of the participants would be unaware of the current and proposed smart technologies and thus began with a brief overview of smart technology development. The participants viewed PowerPoint presentations, including short videos, which introduced all four of the technology categories with a particular emphasis on their use for aging in place (video list, Appendix A). The videos that were chosen demonstrated the ways in which smart technologies could be used to support aging in place while, as far as possible, avoiding social judgments or discussion of ethical issues. The participants were told that the videos were for illustration only.

#### 2.5.4. Connection of Diverse People

The participants comprised people from a range of backgrounds and ages. When changing tables, the participants were encouraged not to remain as a group, so as to hear and respond to the experience and opinions of all of the participants.

#### 2.5.5. Listening Together for Insights, Patterns, and Deeper Questions and Making the Collective Knowledge Visible

As the participants moved, the researchers remained at a table and summarized the previous discussions for each new group. Thus, the discussions built on the issues previously raised and began making the collective knowledge visible. During a break, the researchers collated the tablecloth notes and presented the findings back to the whole group, allowing the participants to provide further feedback on the findings and the World Café process itself.

### 2.6. Analysis of Data

The deliberations were audio-recorded but, except for illustrative quotes, not transcribed. The researchers (JS, HB, LC, FM, AC, LP, KP, TB, BF, RC, PS, AB, SC, KM) undertook the initial coding and generation of themes in the table summaries. The tablecloth notes were transcribed by one author at each site (LC, FM). An initial analysis of the South Australian Cafés was conducted by two authors (HB, FM) for a report [5]. This preliminary summary of the findings provided a foundation for the thematic analysis which incorporated the two NSW Cafés. At this point, each summary was checked against the recording and augmented, as appropriate, by a researcher (JS, LC, AC, LP, KP, TB, BF, RC, PS, KK, AB). One author (JS), in discussion with two of the other authors (JE, SHE), conducted a thematic analysis, drawing together the major themes, extracting relevant quotes, and ensuring consistency across the sites. All of the authors provided feedback on the final selection of themes.

Many of the themes emerging from the data reflected the conceptual framework developed by Beauchamp and Childress [19,20]. This framework is built around four main principles—respect for autonomy, beneficence, non-maleficence, and justice [19,20]—with a set of related concepts including independence, consent, confidentiality, privacy, and equity. These principles remain influential in the evaluation of issues of an ethical significance in the field of health. In this context, the principle of respect for autonomy is particularly pertinent, since there is an inherent tension between assistive technologies promoting independence and autonomy and acting to monitor and secure, thereby potentially leading to obtrusion, intervention, and restrictions [21,22]. As Beauchamp and Childress indicate, understanding is an essential element for “autonomous action” because “an action is not autonomous if the actor does not adequately understand it” (p. 104) [20]. Given the novel and complicated nature of smart technologies, understanding may be an issue in their rollout. The principles of beneficence and non-maleficence obligate those providing care (here, arguably, including technology developers) to support the wellbeing of others; in the context of aging in place, the technologies should be beneficial and not cause harm. We explored the real and potential benefits and harms of smart technologies that the participants identified. Finally, the concept of justice has been interpreted as “fair, equitable and appropriate treatment in light of what is due or owed to persons” (p. 250) [20]. Distributive justice refers to the equitable distribution of benefits and burdens. Given the digital divide highlighted earlier, particular attention to this principle is warranted.

After an initial analysis, the themes were mapped against the principles, while accounting for the framing shift that a focus on big data creates. Here, we drew on the work of Richards and King [23] who examined these concepts in the light of “a new digital society” and the need to develop “big data ethics” (p. 395) [23], as well as their concept of identity as “the ability of individuals to define who they are” (p. 396) [23]. This concept was particularly useful in understanding the participants’ views about how smart technologies (might) shape their lives and current and future social mores, human thought, and expression. Considering this conceptual framework, we re-examined our themes, grouping, recoding, and collapsing codes, and extracting quotes.

## 3. Results

The participants (*n* = 84) were engaged at five sites across two Australian states: two urban (Playford, Noarlunga) and one rural (Port Pirie) in SA, and two in the NSW Illawarra region (Wollongong, Dapto) (Table 1).

The commonly identified themes related to the perceptions of potential benefits and harms associated with the use of smart technologies, such as the benefits of convenience and increased safety versus concerns about data security and social isolation. Other themes related to concerns about the impact of smart technologies on social norms and how we related to each other in community.

Many of the themes were common across the different technologies presented. Acceptance was generally lower for robotics or smart homes than for AVs and smart wearables, although many of the participants held strong concerns about vehicle safety. Themes were also common across the Cafés; however, some reflected the surroundings in which the participants lived. For example, the Port Pirie participants lived in a rural setting with unreliable internet access and wondered if AVs could cope with rugged road conditions and wildlife (kangaroos, donkeys, camels) on the roads.

Below, we describe the participants’ responses across the World Cafés. We acknowledge that these were shaped by the introductory videos, though many drew on previous encounters with smart technologies in their daily life. As the table discussions progressed, and the participants reflected on the known or potential impact of technologies in their own lives, the videos were mentioned less often. We describe the participants’ understanding of ‘smart technologies’, including a degree of acceptance or resistance across the groups, apparently unrelated to the videos but instead reflecting deeper beliefs. The final section focuses on the ethical issues. Occasionally, the participants used words with meaning in ethical frameworks (i.e., confidentiality, privacy); more commonly, they used expressions from which it could be inferred that they held concerns in these areas.

### 3.1. Understanding, Acceptance, and Resistance

Some of the participants had seen or ridden in an AV or had multiple automatic functions in their cars, and many understood the potential applications of wearables because of the widespread use of emergency alert and personal fitness devices. In contrast, many struggled to understand how smart homes and robotics could be successfully integrated into their lives and found it hard to imagine the IOT in real life. The examples helped, but the conversations typically drifted back to mobile phones or social media applications, such as WhatsApp™ or Facebook™. There was widespread confusion about what constituted ‘the internet’. Some participants indicated that they never used the internet, yet on further probing stated that they played games with friends and, occasionally, ordered goods online. One participant expressed distrust of all social media, particularly Facebook™, but regularly used WhatsApp™ to communicate with family. Many, despite owning smart devices, used very few of the functions. Others were not sure exactly what they owned or used.

*A lot of other seniors, that I know, still just have a mobile phone, it is not a smart phone and they can get a bit confused as I was at first as to what’s the difference even*.(Female 1, Noarlunga)

*Trying to understand the internet and what it offers is my biggest problem*.(Female 2, Noarlunga)

Some of the participants raised concerns that were deemed to be major impediments to using smart technologies, often assuming that all human involvement would be removed. For example, one participant was concerned that a service in which volunteer drivers assisted severely disabled people would be unworkable with smart vehicles; another wondered who would look after the people injured in a crash if the ambulances were driverless. Others asked how, in a world of driverless public transport, fare-paying would be monitored, and walking aids and mobility scooters accommodated. Others envisaged robotics as androids only, discounting more prosaic extant robotic versions, such as robotic vacuum cleaners and medicine dispensers. Some voiced concerns that the volume of data fed to health practitioners through health-monitoring wearables and smart homes would be overwhelming and unmanageable for the provider.

The participants’ responses regarding smart technologies ranged from outright rejection to enthusiasm and excitement around the potential for change. The resistance was often based on a dislike of change and the new social mores, and some of the participants could not envisage any circumstances under which they would be willing to use smart technologies. Conversely, many were happy to accept them, provided they were useful and easy to use.

*All the best technology in the world doesn’t help if you’ve got someone who doesn’t want to use it*.(Male, Dapto)


*We use the internet a lot. I accept any new technology. Ya ya, I like it… Makes the life easy.*
(Female, Noarlunga)

The opposition to the introduction of new technologies was expressed as both a lack of interest in new technologies and anger at being forced to participate in unwelcome change.


*In the respite, they brought iPads for everyone to play with and to teach them how to use it and no one was interested. No one wanted to know.*
(Male 1, Noarlunga)


*Why should people, especially of our age, be forced to go online to do something which we have always done with a pen and paper? We are losing our independence to electronic technology.*
(Male 2, Noarlunga)

One participant, talking about a robot which had a screen via which you could see another person, said:

*I’d put a hammer through it, I really would*.(Female, Port Pirie)

Others suggested that it was about taking time to adapt to the new technologies: “*people talk about technology as if they are scared*”, but all of them have devices and “*get used to them*” (Female, Playford). Some of the participants considered resistance to new technology as a perennial problem, whereas others thought the scale and pace of change was unprecedented:

Female 1: *We are probably saying just what our parents and our grandparents said.*

Female 2: *They weren’t faced with the amount of change as what we are facing.*(Port Pirie)

### 3.2. Ethical Issues

#### 3.2.1. Beneficence (Benefits) and Non-Maleficence (Harms)

The participants described many of the benefits associated with the use of smart technologies for those aging in place, including potentially enriched quality of life, increased convenience and safety, and through providing solutions to issues which emerge with aging. However, some of the participants tempered these perspectives by describing potential harms from ill-considered applications of their use, including a loss of social contact, personal and societal cost, job losses, potential vulnerabilities in the technologies, and a general fear that it would make people lazy and purposeless.
*… the more free time I’ve got the more I can do the things I really like to do rather than the things I have to do*.(Male, Dapto)
*You’ll stop thinking, you’ll lose motivation and that’s what worries me*.(Female, Wollongong)

The smart technologies were visualized as providing facilitated access to the arts and new ways of communicating with friends and family:
*I think that really opens up the world…If technology does those sort of things, which are really positive enriching your life, I’m all for it*.(Male, Dapto)

A common concern was the threat of losing their driver’s license in older age, resulting in a loss of independence. Many of the participants saw AVs as a solution to this issue, but many also suggested that relinquishing driving would be shedding a burden:
*I think just ease of lifestyle…Because if you don’t have to drive, so a trip, if it is a longer trip, it’s much more pleasurable because you are not concentrating on driving. You’re not worried about safety…You can all just sit together and enjoy each other’s company.*(Female, Wollongong)

However, others, primarily males, said they enjoyed driving and would miss it if AVs became common:
*I don’t drive a vehicle every day, but I enjoy driving. The big problem I see generally with automatic functions. It’s alright to have the washers go on and your lights go on but it takes the joy out being able to drive …*(Male, Wollongong)

Convenience was seen as a major benefit with AVs. For example, the participants believed that an AV could replace a flat tire or automatically call a second vehicle if it broke down. They suggested that using an AV would reduce the stresses associated with having to drive in heavy traffic, find parking, access scarce public transport, or maintain and store a vehicle:
*You won’t have to own a car and you won’t have to have your garage… you’d get there and you’d get out. Don’t have to worry about parking. I think it’s a fabulous idea.*(Female, Wollongong)

However, one participant questioned whether AVs would be as spontaneous, convenient, and timesaving as being able to just *“jump in my car”* (Male, Port Pirie).

Some of the participants marveled at how useful smart homes would be, in activities including housework, lifting heavy objects, opening cans, controlling water temperature, locking doors, raising blinds, finding lost articles, and in doing all of these things remotely. The participants described how current technologies enriched their lives:

*The microwave. Fantastic. Forty seconds, your breakfast is ready to eat whereas I can remember getting up, lighting the wood fire, waiting for it to burn, waiting for the stove to heat up to warm your milk to make your breakfast. It is very time-consuming*.(Female, Port Pirie)

However, a few mourned the loss of previously routine services, such as morning milk on the doorstep, and thought that smart homes represented a further decline in the quality of life.

The participants also considered how smart technologies might create safer environments, addressing the dangers posed by the loss of skills associated with aging, including the capacity to drive. Often the discussion focused on the potential benefits and harms to loved ones or society in general:


*At the moment anything goes on the road regardless of the sign on the signpost, if driverless cars were actually programmed for a set speed… that would mean that possibly you would feel more safe in a vehicle that you knew was only going to be driving at 80 kms/h and everyone else’s vehicle was also driving [at 80 km/h] so you would have an element of “I trust that we will stop in enough space’s time” whereas now it is random.*
(Female, Dapto)

*I was just thinking of the young ones going home at night, the drunk girls going home in the night, how do you stop the male getting in the car with the drunken girl*.(Female, Wollongong)

In the home, the benefits of increased safety for aging in place were more consistently recognized by participants, including automatically turning off kitchen equipment or alerting a monitoring agency or family member if a fall or adverse event occurred. Some of the participants saw the advent of smart technologies as an opportunity for a new industry protecting the vulnerable, physically and from exploitation and cyber-attack.

*I would adopt some of this technology right now because I live alone...I don’t want to have an alarm, but I would love to be able to ask for assistance if I was incapacitated in my home… I don’t have family… I would adopt it for safety reasons*.(Female, Dapto)

However, some of the participants questioned the trade-offs between safety and privacy. and wondered if it was worthwhile or necessary:


*What freedom do you have when you have this continual oversight and supervision of you in your own home? There’s a balance between having some safety versus having some independence and I think that really usurps your right to privacy.*
(Female, Dapto)

Loneliness and reduced social contact were considered to be the major problems. One participant reflected the experience of many when she talked of visiting others and finding that “*I was the first person the people in those homes had spoken to all week*” (Female, Playford). Many were concerned that the new digital technologies, such as smart phones, had reduced face-to-face social engagement and interaction across society:

Male: *It’s isolated us. It’s put us into silos.*

Female: *We are more isolated than ever now and lonelier.*(Noarlunga)

Some worried that the additional smart technologies might exacerbate these trends by replacing and removing human contact points, whether through a bus driver or carer, or in the displacement of current networks sustaining some communities. The importance of face-to-face human contact was a common theme across the Cafés.

Female 1: *I mean we sit down and we have a laugh and that’s what you really need isn’t it…*

Female 2: *Yes, the communication face to face*.(Playford)

In one Café, the participants spoke extensively about how their neighbors and families had routines for looking after each other, ranging from regular phone calls, sharing house keys, and informing others of temporary absences, through to using the opening and closing of curtains to signal wellbeing. These routines helped people to retain regular contact with friends and family, and, whilst recognizing that smart home devices might be more reliable and responsive in emergency situations, some feared this contact might cease if these neighborly actions were overtaken by smart technologies.


*We’ll just put Nana in a house with a robot and stuff and we’ll come and see her next year.*
(Female, Wollongong)

The participants clearly distinguished between the robots as ‘partners with human carers’ and ‘replacements for human carers’. Thus, whilst recognizing the value of the smart technologies for providing emergency communication, linking people to support systems, and as an aid for times when human carers were absent, the participants were very reluctant to replace human carers completely. Across the Cafés, many held that face-to-face human interaction in caring was irreplaceable and that the smart technologies should only be used to augment and enrich the services provided by human carers:

*If the robots are doing the jobs, perhaps there is employment for people coming and doing the socializing*.(Female, Dapto)

Using a robot for conversation or companionship was not seen as useful:


*You know that it’s automated. You know it is not a person. It’s like talking to a machine on Telstra, you know they are not listening to you.*
(Female, Port Pirie)

The two exceptions were for dementia patients, and those who did not choose to engage with other available people:

*…dementia patients, they repeat themselves a lot and that’s where maybe, hey, like with the right program was in place…because all they want is somebody to listen*.(Female, Port Pirie)

*When I first watched that lady dancing with the robot [in provided video], a part of me was a bit sad, but then again I’ve managed eight retirement villages and there are some people who are so lonely and no matter what you do to try to involve them in the village community they don’t want to be part of it, so perhaps for some people that is the only way that they can have some sort of communication*.(Female, Wollongong)

#### 3.2.2. Respect for Autonomy—Identity, Independence, and Self-Determination

The participants described various ways in which smart technologies might impact on their identities and, more broadly, on the nature of the society in which they lived. For example, the participants were worried that technological change would impact on people’s ability to think and innovate and that it would encourage laziness, selfishness, and reduce empathy and compassion.

*It makes a person more selfish as you are asking a robot all the time about your needs, what you need, what you want and what you want to know, whereas when you have to interact with another human being you’ve got to fit in with their moods and sometimes I might come in really happy and I want to talk about what happened last night at a party or whatever and I see my friend and she’s got some issues at home and so I have to tune into that*.(Female, Playford)

Some of the participants challenged the assumptions that older people must be cognitively or physically slow, and that this was necessarily a bad thing.

*I’m old, I’m not gaga*.(Female, Port Pirie)

Another recounted being asked on a yearly basis, “*when did I last fall over*?”, and her initial response had been, “*I can’t remember*”. This revelation was met with laughter, as the other participants thought that this was open to misinterpretation. One person said: “*You never say that to a doctor when you get older, never”*. Another corroborated the assumptions made about older people, saying:

*But this is what you learn when you get older as they are so happy to slap the [negative] name tag on you*.(Female, Port Pirie)

The participants’ responses also reflected their distress about their potentially limited capacity to successfully live in this changed society. In particular, maintaining independence was seen as an unquestioned right which may or may not be supported by the new smart technologies; many spoke of relatives or community members who were hugely impacted when they lost their ability to drive. One participant said that, although it might have been difficult for his father to accept, he probably would have used driverless cars:

*I think to get his independence back. When he lost his licence, he lost his will to live. He lost his independence…I saw him deteriorate very quickly when he lost his licence*.(Male, Wollongong)

Another participant, reflecting the views of many, described her personal motivation to learn how to use AVs:

*If I had a choice between sitting at home or being able to get into an autonomous vehicle, I’d go with the autonomous vehicle. I would absolutely do what do I need to learn because I would really want to have that independence and that would be such a motivator*.(Female, Dapto)

AVs were also seen as a boon, in that they would increase autonomy and choice for people with limited mobility.


*…this would encourage a lot more flexibility being able to go out particularly if they have got a carer so that it is not only just them, it is them and their carer could actually go to so and so and have a coffee, go to the movies because at the moment they are stuck with community transport, with whatever private arrangements are in place, so it gives them a lot more flexibility.*
(Female, Dapto)

In the case of smart homes and robotics, some of the participants were concerned that robots would be “*bossy*”, telling them: “*you take your medication, you do this, you do that*” (Female, Noarlunga). “*Doing things for yourself*” was seen as important for keeping you going and living longer:

*If it is going to do everything for you, I’ve lost my independence*.(Female, Playford)

Across the groups, the participants valued self-determination and particularly feared loss of control to robots.

Female 1: *I think we’re going to become the robots and they’re going to be the intelligence. We need to think.*

Female 2: *Yeah.*

Female 3: *We’re giving [away] too much of our control*.(Port Pirie)

For others, the fears were related to the potential for an external entity to assert control:


*It is more like controlling, I worry about us being controlled by the government. Will it take your right to own a car away?*
(Female, Noarlunga)

Some of the participants were concerned that engagement with these technologies might fundamentally change who we are as people:

Female1: *See the more and more we talk about all these different things, it is almost like we’re becoming robotized—is that a word, robotized? [general laughter]*

Female 2: *desensitized*

Male 1: *yeah, desensit…dehumanized. That’s the word, exactly.*(Port Pirie)

However, some acknowledged their current use of automated items in cars, including lights, automatic transmission, braking, and windshield wipers. In this light, the participants agreed that some control by externalities would be acceptable within boundaries, including the potential to de-activate a robot or override dangerous autonomous driving, and that monitoring through the IOT would be acceptable with their consent and when conducted by a family member or trusted care provider.

#### 3.2.3. Privacy and Confidentiality

Many of the participants across the Cafés were concerned about maintaining privacy. Some of the participants asked if it would it be possible to do things without detection by an in-house robot, seeing this as a potential intrusion into their personal space:

*There are some things that you don’t want other people to know about you*.(Female, Playford)


*How much will it intrude? If you don’t want the robot to know something, can you turn it off?*
(Male, Playford)

The participants were divided about whether robotics would offer more or less privacy than human carers. This dilemma was a ‘*double-edged sword’*, in that personal contact from a human carer was seen as good but also invading privacy. For example, when showering, a robot “*would be less embarrassing wouldn’t it?”* On the other hand, some sought reassurance about limited access to sensitive, private information:

Female: *Who would control the robot while they are washing you?*

Facilitator: *It would be a machine learning robot so it would have learned how to wash you.*

Female: *So there would be no contact with the outside world?*

Facilitator: *No*

Female: *That’s alright then.*(Playford)

Some observed that surveillance was already ubiquitous, so any concern was futile, and posited that the intrusion of surveillance would be acceptable if they could stay in their own homes. Others identified surveillance as indicative of an insidious and over-controlling authority.

Female 1: *The systems within these cars would be feeding back all the information about you back to some central area.*

Female 2: *It’s getting a bit big brother.*(Port Pirie)


*I’m nearly 85 and I’d be prepared to have things more intrusive to be able to stay where I am…I still wouldn’t like it but I’d be prepared to do it.*
(Male, Wollongong)

The participants had explicit concerns about confidentiality, holding that the information accessible via smart technologies was very sensitive:


*What about confidentiality? Who has access to this person’s data? You know, if they are taking their medication, have they fallen, are they having a depressive state, are they contemplating suicide?*
(Female, Playford)

Overall, the participants simply did not want anyone having knowledge about particular things they deemed to be private, but appeared less worried about the confidentiality of any collected information. Some observed that breaches in confidentiality were not confined to the smart technology space:

Female 1: *I don’t know because it depends. What happens if you lose control of your bowels and you have to be cleaned up, you don’t want your carer to go and tell the next-door neighbor.*

Female 2: *They are not supposed to go and tell the neighbor are they?*

Female 3: *But they do.*(Playford)

The concerns about confidentiality and privacy in association with AVs and the IOT often incorporated fears about the security of computer-controlled systems, cyber-crime, cyber-terrorism, and the perceived ease with which outsiders might be able to “hack” into systems controlling the AVs and the IOT.


*If someone hacked into it, someone could take control and take you off.*
(Female, Dapto)

People sometimes drew on their experiences with financial fraud and scamming, as well as reported vulnerabilities in broader systems, fears about malevolent foreign entities, and perceptions of over-reliance on the internet and power-based systems.


*When the computers go down at the banks, they can’t function…you can’t get your money out.*
(Female, Dapto)

Female 1: *It’s too easy for people to scam you.*

Female 2: *And it’s scary.*

Male 1: *It’s scary when you call up a company and you end up talking to someone in the Philippines or maybe in India and they want your credit card so they can get payment and you wonder where is this going?*(Noarlunga)


*All of these technologies revolve around power so if you want to destroy a country, if you want to shut something down, you shut down the power and in a few days you can’t recharge anything.*
(Male, Dapto)

A common sentiment was that:

*Until someone can convince us that it is going to be safe why should we let you have too much of our information*.(Male, Port Pirie)

#### 3.2.4. Justice and Access

The participants across the Cafés indicated that equitable access to the technologies may be hampered by cost and an individual’s capacity to source, set up, maintain, and use the technologies effectively. The affordability of the technologies and the underpinning essential internet access was identified as a major issue.

*I’m on the pension, I guess most of us here in this room are on the pension and I mean every day you think about the cost of things and so yes it would be lovely to say ‘yes I would like one of these vacuum cleaners and whatever that would go around and help me’… but I can’t afford it*.(Female, Noarlunga)

For others, the new technologies offered opportunities that their current financial situation might preclude or make difficult. For example, one participant who could not afford to replace his aging car indicated that access to an AV might overcome the problems this engendered.

Facilitator: *If you could get an autonomous bus to come to your door?*

Male: *It would be ideal. It would answer the whole problem, a driverless bus*.(Noarlunga)

Some of the participants indicated that smart technologies may be particularly useful in improving access to services. For example, the participants envisaged that the AVs could provide accessible point-to-point transport and that smart homes could support aging in place. The Playford participants described how an autonomous bus, which transported people from the station to the local hospital, allowed the passengers to alight directly outside the hospital (especially salient for older people with mobility issues), thus avoiding worries about organizing, paying for, or overstaying expensive parking. Many of the participants recognized the value of smart technologies in helping frail people to be cared for effectively in their own homes.

*Could a robot shower you? I mean you can’t always get the help. You could have it when you wanted it not just when they [i.e., carers] could come*.(Female, Wollongong)

A few of the participants indicated that smart technologies may be particularly useful in meeting the needs of the aged population, especially for people with disabilities, such as cognitive impairment, mobility constraints, vision and hearing loss, sleep disruption and depression, and impairments which result in skill gaps.


*I can’t read very well, but having a robot read for me would be very good.*
(Male, Dapto)

The Dapto participants asked about the potential for smart technologies to improve hearing aids and solve age-related issues with tinnitus, macular degeneration, anxiety, and depression. Some valued the idea of robotic pets for people with dementia; others hoped that smart technologies could detect problems early and initiate responses before the problems became entrenched. However, many of the groups also observed that using smart technologies might exacerbate cognitive and physical decline:

Male: *That’s another thing, you don’t want to make people useless*.

Female: *You’re going to get Alzheimer’s quicker*.

Female: *Probably die quicker*.(Playford)

The cost of installation, maintenance, electricity, and the remote monitoring of wearables and home systems to support aging in place were also sources of concern:

Female 1: *Your house would have to be wired for all these sensors …which would be a massive job and cost.*

Male: *and who’s supposed to foot the bill for that?*

Female 1*: And if it’s monitored somebody’s got to be monitoring it. You’d have to pay for that service, I would say*.(Port Pirie)

Some of the participants were more positive, pointing out that invariably new technologies were expensive at first but then the prices dropped, increasing consumer access. However, the Noarlunga participants noted additional complexities, in that individuals would need a level of technology ownership before they could use some of the smart technologies. For example, the use of AVs would require a smart phone with an app and access to a credit system, effectively excluding many older people.

The issues of access were not limited to the financial costs. The participants across the groups described the challenges they faced in trying to install and use smart technologies, often relying on younger family members:

*…I have always relied on my children to activate the technologies and then hand them over to me and say: ‘all you do is press this button’…The older you get the less inclined you are to engage with things like technology*.(Female, Dapto)

Many of the participants acknowledged that, with aging, it may become more difficult to learn and remember how to use technology, and that extra effort may be needed to ensure that the “*brain [gets] used to new things*”. This was best achieved by “*keeping things familiar and then gradually change things*”, so that not everything was “*happening overnight*” (Playford).


*You know someone comes along and says: ’This is how it works. This is all you have to do.’ That goes into your short-term memory…But if you start to have memory loss which is pretty on the cards for most of us, how are you actually going to have enough successful activations to commit it to long-term memory.*
(Female, Dapto)

Other participants thought that a technology assistance robot might be more useful than family members or technicians, who often talked a language that they did not understand, gave incomplete knowledge, or talked too fast. The use of acronyms was singled out as a particular challenge.


*The language they use for the technology, the acronyms that come in, I’ve no idea what they mean [another man laughs] and well they say ‘if you want this, do this’ and I’m like ‘what?’. ‘Look at A, B, C, D.’ You know, I have no idea. I had to ring me son.*
(Male, Playford)

## 4. Discussion

The aim of this study was to collect older people’s perspectives and preferences on the application of smart technologies for aging in place. We found that many of the participants had little understanding of how smart technologies might support them in staying at home and had experienced difficulty using digital technologies. These findings support other studies reporting participants’ lack of understanding about smart technologies [24,25,26], where they question their competence in managing the technical aspects of setting up or operating the devices [26,27,28,29], or describe the large burden of effort, time, and patience required to keep up with new and rapidly changing technologies [8,29,30]. Understanding, according to Beauchamp and Childress [20], is an essential element for informed consent and autonomous action—although they warn that restricting “adequate decision making…to the ideal of full or completely autonomous decision making strips their acts of any meaningful place in the practical world, where people’s actions are rarely, if ever, fully autonomous” (p. 104) [20]. Understanding relates to both access to information, which can form the basis of informed consent, and the capacity or competence to understand that information (p. 124) [20]. Amongst the participants, AVs and smart wearables were more familiar and perhaps, therefore, more widely accepted than the less familiar smart homes and robotics. Ghorayeb et al. [25] and others [8,31,32] suggest that familiarity both with the technology and the providers can build trust and confidence, thereby increasing acceptance of smart technologies.

As others report [8,24,25,27,28,33], some people rejected the use of any technologies requiring internet access, while others enthusiastically welcomed the advent of new smart technologies. Most, however, exhibited a cautious acceptance of smart technologies but with caveats, accepting their use only under certain conditions, particularly as an alternative to institutional aged care and with the additional stipulation that smart technologies should augment the actions of human carers and not simply replace them. As reported elsewhere, some were willing to accept assistive smart technologies reluctantly for “peace of mind” (p. 92) [34], or because it could reduce the burden on others [8,12,25,27] or improve their personal safety [27,29,35,36].

These findings suggest that the resistance to use of smart technologies, at least in part, may be founded on a lack of familiarity and/or difficulty with digital technologies. Birkland suggests strong resistance in some people may arise from “extremely traumatic experiences with technology in their early to mid-adulthood” [33]. Individuals may become discouraged if they feel that the technology is too complex for their needs [12], or does not suit their daily routines [8]. Many of the participants described their struggles using new technologies and their fears that inexperience may open them to exposure of their personal information. This is a reasonable concern since there is evidence that older people are at higher risk of being the target of scams [37,38]. Research from Australia, demonstrating a digital divide [13,14,39], reflects similar findings from other countries: for example, although many US seniors have access to smart phones (42%), and half have broadband internet access at home, many remain disconnected from online information [40]. Similar to Australian data, use drops with increased age and is lowest in the most disadvantaged households [40]. Even amongst those using electronic devices for online access, confidence is much lower than for adults aged under 65 [40]. The gradual introduction of smart technologies into homes [8,25] may mitigate some of the issues and the problem may diminish as age groups more familiar with digital technologies enter this space. Our participants suggested that an assistive robot helping them manage the network of new technologies might be helpful, but an alternate solution would be to incorporate social interaction with a human technician [8] or a peer leader [12], who could offer both education and technical assistance. This would have the added benefit of building trust and confidence in using the technology.

The principles of beneficence and non-maleficence [20] suggest that the designers of smart technologies and the policy-makers supporting their adoption should strive to prevent or reduce harm to users and that, where harm is unavoidable, the benefits accrued should greatly outweigh any harms. Our findings, in common with many other studies [25,27,35], indicate that the participants believed that smart technologies may provide mixed outcomes and brought both potential benefits and harms.

A widely lauded benefit of smart technologies amongst our participants was the use of AVs to support independence and mobility, should they lose their driver’s license. This benefit is little reported in literature reviews of community views on smart aging [27,29,35,36,41], as these primarily focus on smart homes and wearables, but it is noted in papers focusing only on AVs [42,43]. Hagzare et al. (p. 2) [43] suggest that acceptance by older people relates to the balance between “the perceived reliability of the system and the perceived benefits of the technology” and that, in participants with exposure to AVs, coherence between their driving style and the driving style of the AV (e.g., aggressive or cautious) may impact on their acceptance. This is in line with our findings that technologies should be individually tailored, rather than ‘one size fits all’. However, some views may reflect the real risks posed: some critics suggest that AVs may actually increase risks for older people in the transition period from partial to full automation [44].

In common with other studies [27,29,35,36], our participants recognized the benefits of assistive smart technologies in the home. There were some differences: although earlier studies reported positive views on the impact of smart technology on social interaction, our findings echoed UK [25] and US [12] research, wherein many of the participants were concerned that the technologies would exacerbate social isolation. Specifically, the participants highly valued face-to-face interactions with service providers, friends, and family. An international Pew Research Center survey indicated that many people hold negative views on the replacement of human labor with robots [45]. These findings suggest that although safety is important, particular attention should be paid to promoting independent but socially connected lives.

The participants in our study expressed concern that they might be stigmatized if they used smart devices designed for monitoring or assisting older people. This is similar to views expressed in other studies [25,27,34]. Ghorayeb et al. (p. 10) [25] suggest that smart technologies designed for the general public serve to “increase confidence …and helps [older people] feel more rather than less capable. Conversely, technology explicitly described as for the benefit of older people is seen as less acceptable”. As smart homes and AVs become more common in the general population, this distinction may dissipate, but it underlines the need for developers and care providers to be mindful of this issue.

Interaction between designers, service providers, and end-users, before and during the development and roll out of technologies, could help to promote the benefits and mitigate harms. In a 2009 study, up to one-third of assistive technologies were abandoned within one year of use [11]. The individuals withdraw from use for multiple reasons but a disjuncture between the expectations and beliefs of developers and those of end-users may be a major contributory factor. In a 2020 study in Australia, trialing the use of ‘smart home’ technologies, some households withdrew from the project citing feeling “overwhelmed with the devices”, or that a partner’s condition (e.g., dementia) made the presence of devices too stressful (p. 110) [8]. Many smart technologies are envisaged by young designers, who may have little understanding or input from older end-users [43,44]. In future research, bringing the technology developers together with the service providers and end-users in interactive discussion would help delineate the areas of difference and similarity in the views and objectives of the groups, and thereby support improved designs. We discuss an example of this disjuncture in held values in the paragraph below.

The terms ‘privacy’ and ‘confidentiality’, although commonly used interchangeably, are related but different concepts. Privacy refers to the right to control access to oneself, including information about oneself; confidentiality relates to the information only, so that if a person holds information about another in a “confidential relationship”, they will not share that information without permission (p. 316) [20]. This applies to many service providers who hold information about patients but are bound not to disclose it. However, the technology development researchers, in a study by Birchley et al., “predominantly characterize(d) privacy as the unbidden sharing of data” (i.e., the concept traditionally characterized as a breach of confidentiality) rather than as “an invasion of personal space” (p. 4) [46]. In particular, some of the researchers thought that “loss of privacy is a ‘done deal’ and therefore not an important worry” (p. 4) [46]. This interpretation is at odds with the findings from our research, where many of the participants, in line with Beauchamp and Childress [20], understood privacy as the right to control access to oneself and were concerned about its loss. Our participants were also concerned, but less so, with breaches in confidentiality, particularly the vulnerabilities in interconnected e-systems to malicious use. The concerns about privacy and confidentiality are widespread [25,27,29,31,36,47]. The technocratic worldview of privacy advocated by the researchers in Birchley et al. [46] is challenged by Richards and King (p. 395) [23], who argue that “we must recognize ‘privacy’ as information rules”, and that “while the amount of personal information that is being recorded is certainly increasing, so too is the need for rules to govern this social transformation”. Our research suggests that institutionally supported boundaries would enhance trust and increase the acceptance of smart technologies amongst older people.

Being autonomous is usually taken to mean that people can make their own decisions and thereby shape and direct their own lives. Within the concept of respect for autonomy lie notions of independence, self-determination, privacy, identity, and confidentiality, here including the specific concerns about the vulnerability of technologies and informed consent. A duty to respect autonomy encompasses a duty to support people to make their own decisions, free from coercion and stigma. Beauchamp and Childress [20] focus on what they describe as “non-ideal” rather than “ideal” conditions and describe autonomous action as dependent on three factors, the first being intentionality, where a person may plan and execute actions which they may or may not wish to perform. This includes circumstances where they have weighed the options and chosen a path which often “reflects conflicting wants and desires”. This was apparent in many of the participants in our research who described resisting the use of smart technologies in a variety of ways, but who also indicated probable acceptance, as they aged, if the technologies supported their continued independence and their associated usual activities.

The research described here confirmed the previous international studies that the predicted capacity to remain independent was a major factor in the acceptance of smart technologies in the home [8,27,29,35,36,48,49]. However, even where some dependency occurs, self-determination is still possible [8,50]. As Remmers suggests, humans must at some points in their lifetimes be dependent on the care and support of others and these “reciprocal dependencies” are a key component of a functioning society (p. 203) [50]. When people age, they are increasingly likely to require some form of dependent relationship, but this does not preclude living a self-determined life. Remmers calls for “facilitating the independence of older people” to be seen as “a social function, which affects everyone’s given right to freedom and self-determination” and that ”the right to self-determination is an obligation in the sense of self-responsibility” for the older person (p. 204) [50]. It is of interest that the participants in our study did not directly discuss their freedom to take risks [21], though this was often implied in the discussions. Wareham (p. 129) [51] argues that we need a better recognition of the scope of the ethics of aging and, particularly, that we should not see “aging persons as objects of ethical dilemmas and policy”, since this frames the elderly as ‘problematic’. Rather, aging should be seen as a whole-of-life process since many of the dilemmas confronted in mid-life are related to aging. Therefore, “the ethics of aging should place the aging person at the centre”, who should be seen as “moral agents or subjects rather than problematic objects in the discourse” (p. 129) [51].

Richards and King suggest that we think about identity not as a “specific name for a specific person” or as “those properties or qualities” which make a particular person, but as “the right to define who I am” (p. 423) [23]. Drawing on the work of Marshall McLuhan, they propose that “as consumers, our identities are increasingly being shaped by big data inference and the companies that control them” (p. 424) [23], and that identities may be compromised “by allowing institutional surveillance to identify, categorize, modulate and even determine who we are” (p. 396) [23]. Our participants frequently expressed concern about the potential adverse impact of smart technologies on individuals and society. This included encouraging selfish and insensitive behaviors, the insidious impact of widespread surveillance, and the potential for increased categorizing and stigmatizing of the elderly. An Australian study on ‘smart homes’ similarly found that some of the participants were concerned that a reliance on ‘smart’ devices encouraged sedentary behavior and they would “lose the exercise benefits associated with everyday activities such as getting up to turn a light or kettle on or off” (p. 64) [8]. This focus on adverse societal impacts may be specific to Australian society. Certainly, this finding is not well explored or identified in the studies from other countries, perhaps because many of the studies have been underpinned by classical technology acceptance models, with a focus on individual acceptance [26,27,52]. It is also possible that some researchers discount certain views because they see them as a generalized resistance to change with increasing age [53]. However, Birkland (p. 73) [33], in discussing views on the use of information and communication technology, describes people with these attitudes as ‘technological resistance fighters’, in that they view technologies through the prism of their “potential to cause moral decay”. In this view, technologies “are not seen as negative, but rather seen as enabling individuals to wallow in negative traits we all possess: laziness, gluttony, waste, and self-isolation” (p. 73) [33].

In considering distributive justice with respect to the use of smart technologies for aging in place, we can consider who will have reliable and equitable access to the internet [8], and whether those who are most vulnerable and marginalized (such as those with a disability in vision, hearing, mobility, or cognitive impairment) will be able to access these technologies. Many of the participants in our study expressed fears about their ability to access and use new technologies founded on the probable cost of the technologies and their personal capability to use the technologies—a major theme in many reviews of older adults’ perspectives on the adoption of smart technologies [8,26,27].

Overlaying the participants’ personal concerns was an uneasiness about the potential impact of smart technology use on society as a whole. In particular, the participants feared that smart technologies might be used as a substitute for human care, as they had in other areas of life, and that this may further exacerbate the isolation experienced by many older people. The participants across the Cafés were adamant that smart technologies should be an adjunct to current human carers, not a replacement for them. This is in line with findings from other studies [12,24,25,26,54]. For example, Jaschinski and Allouch [54], citing their own and other studies, concluded that older research participants believed “that AAL [Ambient Assisted Living] technologies cannot and should not replace human assistance and human interaction, but should be used as a supplement to human care”. The research from the COVID-19 pandemic supports the positive value of face-to-face communication with family and friends, as opposed to digital communication, particularly for older people who live alone [55,56]. However, the work of Strengers et al. [8] suggests that, for those with these technologies in place, the COVID-19 crisis provided new adopters with the opportunity and perhaps motivation to learn how to use digital video communication tools and troubleshoot technical difficulties with online support. This is consistent with our findings, as many of the participants believed that smart communication strategies could be used to “facilitate and create new opportunities” for social contact (p. 8) [25].

The study limitations include that this is an exploratory study focused on the views of older people and did not include the views of carers or families, nor did it tease out completely the differences in attitudes due to age. More research is needed to understand the underpinning motivations for attitudes towards, and expectations of, smart assistive technologies for aging in place.

## 5. Conclusions

To paraphrase Richards and King (p. 395) [23], “big data societies” adopting smart technologies will need to build new systems of aged care and “the values we build or fail to build into our new digital structures will define us”. Our and others’ findings indicate that the values held by older citizens facing the rewards and penalties of aging may be different to those held by their carers, or much younger, digitally-savvy developers. Further, our findings suggest that the developers and implementers of smart technologies should not underestimate the tech-savviness of older citizens, but should work with users to facilitate ease of use and address concerns about privacy and confidentiality. Interestingly, the participants in our study did not see technology as an (external) labor saving tool, but as a (self) enabling tool, and approaching technological solutions for common problems from that perspective may be important for the final acceptance and usefulness of technologies. The findings presented here indicate that technologies for smart aging should be affordable, customized, and must not replace human contact and interaction. Similarly, although safety is important, the emphasis should be placed on supporting independent socially connected lives. As technologies are developed and implemented within existing care systems, attention to understanding and incorporating older citizens’ values will be important for the acceptance and usefulness of these technologies for independent and full lives as we age.

## Figures and Tables

**Table 1 ijerph-19-07817-t001:** Description of World Cafés and participants.

Place Date	Venue	Number of Participants (Male/Female)	Age Range
Noarlunga (SA) May 2019	Council chambers	15 (5/10)	Not collected
Port Pirie (SA) July 2019	Local public library	8 (1/7)	55–60: 0 61–70: 2 71–80: 2 81–90: 4 >90: 0
Playford (SA) July 2019	Community center	18 (2/16)	55–60: 2 61–70: 2 71–80: 8 81–90: 5 >90: 1
Wollongong (NSW) November 2019	Ex-servicemen’s club	23 (6/17)	55–60: 1 61–70: 5 71–80: 8 81–90: 7 >90: 2
Dapto (NSW) November 2019	Community center	20 (9/11)	55–60: 2 61–70: 8 71–80: 7 81–90: 3 >90: 0

## Data Availability

Not applicable.

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
