# Peer review of "Older Adults’ Perspectives of Smart Technologies to Support Aging at Home: Insights from Five World Café Forums"

_ijerph, 2022, doi:10.3390/ijerph19137817_

Round 1
Reviewer 1 Report
I think that the use of technology is essential to help older adults live at home and have a satisfying life as much as possible. I agree with the author’s opinion that in order for the older adults to lead a digital daily life smoothly, we need to know older adult's understanding and values of technology. However, I think there are some concerns about publishing this study.
How much did the video about smart technology shown to the participants affect the result?
Could it have given the potential bias?
About data analysis. Only one author extracted and coded the recorded data. The data was enormous, but is the validity of the analysis results ensured?
I think it will be easier to understand by clarifying the method and process of aggregating data such the opinions, ideas, and values expressed through dialogue and reflecting it in the results.
Also, the concept frame was said to reflect the Beauchamp and Childress citations 19 and 20. It was difficult for me to understand the rationale for applying the extracted results to the four main principles.
Author Response
We would like to thank the reviewers for their useful feedback which we believe has enhanced the usefulness of the paper. We have provided the reviewer's comments in bold followed by our response.
I think that the use of technology is essential to help older adults live at home and have a satisfying life as much as possible. I agree with the author’s opinion that in order for the older adults to lead a digital daily life smoothly, we need to know older adult's understanding and values of technology. However, I think there are some concerns about publishing this study.
How much did the video about smart technology shown to the participants affect the result? Could it have given the potential bias?
This is a limitation with all deliberative work and the focus in using this method is to provide background information to the participants which is balanced and sufficient for the participants to engage with the topic and the research question. Asking participants to discuss the advantages and challenges, presented by smart technologies for ageing in place, would be difficult if the participants had not previously encountered these items. However, we do understand that there may be some concerns about the content of the presentations which is why we included a supplementary file of the video list and acknowledged the issue at lines 228-229. In response to the reviewer’s concerns, we have also added a sentence at line 163 as follows: The videos chosen demonstrated the ways in which smart technologies could be used to support aging in place while, as far as possible, avoiding social judgements or discussion of ethical issues. Participants were told that they were for illustration only.”
At line 230 we have added the sentence: “As table discussions progressed and participants reflected on the known or potential impact of technologies in their own lives the videos were mentioned less often.”
About data analysis. Only one author extracted and coded the recorded data. The data was enormous, but is the validity of the analysis results ensured? I think it will be easier to understand by clarifying the method and process of aggregating data such the opinions, ideas, and values expressed through dialogue and reflecting it in the results.
We appreciate the concerns of the reviewer about the validity of the analysis. In response, we have adjusted the paragraph at line 176 so that it now reads: “The deliberations were audio-recorded but, except for illustrative quotes, not transcribed. Researchers (JS, HB, LC, FM, AC, LP, KP, TB, BF, RC, PS, AB, SC, KM) undertook initial coding and generation of themes in the table summaries. These summaries were presented at the close of each Café and the Café participants provided feedback. The tablecloth notes were transcribed by one author at each site (LC, FM). Initial analysis of the South Australian Cafés was conducted by two authors (HB, FM) for a report [5]. This preliminary summary of findings provided a foundation for the final thematic analysis which incorporated the two NSW Cafés. At this point, each summary was checked against the recording and augmented as appropriate by a researcher (JS, LC, AC, LP, KP, TB, BF, RC, PS, KK, AB). One author (JS), in consultation with two authors (JE, SHE), conducted a thematic analysis drawing together the major themes, extracting relevant quotes and ensuring consistency across the sites. All authors provided feedback on the final selection of themes
Also, the concept frame was said to reflect the Beauchamp and Childress citations 19 and 20. It was difficult for me to understand the rationale for applying the extracted results to the four main principles.
The Beauchamp and Childress framework is a common device through which to view findings which are ethically contentious particularly in the area of health and social issues and has been used previously for the analysis of other areas of technology. For example, see: van de Poel, I. An Ethical Framework for Evaluating Experimental Technology. Sci Eng Ethics. 2016; 22: 667–686. Published online 2015 Nov 14. doi: 10.1007/s11948-015-9724-3
In the absence of a published overarching theoretical framework, which elucidated the ethical concerns in the area, it made sense to use a more fundamental framework. The early thematic analysis, examination of published literature in the area and discussion amongst the research team cemented our choice. We did think that the concept of identity was inadequately covered by the B&C framework and therefore we extended the framework to include the work of Richards and King.
Reviewer 2 Report
I acknowledge the efforts of Street J et al to study the perspectives of smart technology in this older adult target group. Indeed if smart technology is the future for living well and healthy one of the first groups to study is the older population since their living circumstances and health will be impacted first in the near future compared to other generations.
Street J et al used a very nice technique with the world cafe forums to discover the pro's and con's of related to different themes in the target group towards using these techniques. The reported insights could turn out to be very useful to provide direction for future innovation, acceptance and implementation of the smart tech tools, though findings needs to be confirmed in follow up studies
Some considerations:
Power of the study, how representative is this cohort of participants? Taking into account the numbers highlighted in the introduction on users and non users of the internet (line 74-80).
In general the lack of demographic insight of the study population, social economic class, education background, living conditions, gender, culture etc
Furthermore there is a huge age range, while you can imagine different views towards the tools according different age ranges. Therefore the results are a nice starting description of insight but more conformational studies are needed.
These points need to be described / discussed to place the results in the proper perspective and give insight in how generally applicable the findings would be for the older population.
Author Response
We would like to thank the reviewers for their feedback which we believe has enhanced the usefulness of the paper. Below we have provided the reviewer's comments in bold followed by our response.
I acknowledge the efforts of Street J et al to study the perspectives of smart technology in this older adult target group. Indeed if smart technology is the future for living well and healthy one of the first groups to study is the older population since their living circumstances and health will be impacted first in the near future compared to other generations.
Street J et al used a very nice technique with the world cafe forums to discover the pro's and con's of related to different themes in the target group towards using these techniques. The reported insights could turn out to be very useful to provide direction for future innovation, acceptance and implementation of the smart tech tools, though findings needs to be confirmed in follow up studies
Some considerations:
Power of the study, how representative is this cohort of participants? Taking into account the numbers highlighted in the introduction on users and non users of the internet (line 74-80).
As reviewer 3 has indicated, the number of participants and sites is large for a qualitative study. It would be unusual to talk about the power of a study in qualitative or deliberative research. The aim of the deliberative democratic research, and in particular the World Café method, is not to be “representative” but rather to include as diverse a group as possible and to ensure that marginalised 'voices' are included – see “connection of diverse people” at line 164 and in the recruitment strategy at line 105. This was achieved by the wide range of recruitment methods used and the diversity of the sites selected. Diversity exposes participants to a range of views and experience and encourages rich discussion and debate.
In general the lack of demographic insight of the study population, social economic class, education background, living conditions, gender, culture etc.
We did not aim to bring together a particular mix of educational background, gender, culture etc. This is in line with the World Café method. We aimed to maximise diversity by using diverse recruitment approaches and through the choice of the areas where the cafes were held. We would note that all three of the South Australian cafes were held in low-income culturally-distinct areas and that the Illawarra is a predominantly ‘blue collar’ industrial area but also includes a university as a major employer.
In response to the two points above, we have added the following sentence to the recruitment strategy at line 108:
“In both recruitment approaches and selection of sites, we aimed for diversity but also purposely recruited participants from low socioeconomic backgrounds since these individuals, in general, have less input into public policy and fewer choices than individuals from high socioeconomic groups.”
Furthermore there is a huge age range, while you can imagine different views towards the tools according different age ranges. Therefore the results are a nice starting description of insight but more conformational studies are needed.
In the World Café method, and deliberative methods in general, the aim is to purposely recruit people with a range of backgrounds and ages to come together to discuss the issues through sharing their broad experience and views. This encourages rich discussion and enhances the findings.
These points need to be described / discussed to place the results in the proper perspective and give insight in how generally applicable the findings would be for the older population.
General applicability is not an aim in qualitative research or the World Café method. The aim is to provide a rich discussion of the views of a diverse group of people drawn from the population. We have also addressed this issue in the added paragraph on the limitations at the end of the discussion as follows: "The study limitations include that this is an exploratory study focused on the views of older people and did not include the views of carers or families. Neither did it tease out completely the differences in attitudes due to age. More research is needed to understand the underpinning motivations for attitudes towards, and expectations of, smart assistive technologies for aging in place."
Reviewer 3 Report
The manuscript is both current and topical exploring the uncharted territory on older adult’s smart technology use predispositions and experiences, projecting future adoption use. This was a qualitative size of a good sample size. Use of the World Café form for data collection was most appropriate. Findings are meaning and thoughtful. The discussion of findings is informative. Improvements are possible to add a section on the theoretical foundations in the: value based social science and deliberative democracy to the introduction and framing the study on these key concepts rather than jump them in under methods and materials. Also, the introduction section would be better with questions for study towards the end and just before the methods section so readers can anticipate the answers under Findings. Much more important, the Discussion section is way to long and quite repetitious some places. Authors should cut this down to by at least 2 pages. Some of that wring space “saved” should be used for a section on implications for research and practice (1/2 page) and another section on Limitations of the study and suggestions for further research (1/2 page). The conclusion needs revisiting to state the key findings in 5-7 sentences, with a statement of implication.
Author Response
We would like to thank the reviewers for their feedback which we believe has enhanced the usefulness of the paper. Below we have included the reviewer's comments in bold followed by our response.
The manuscript is both current and topical exploring the uncharted territory on older adult’s smart technology use predispositions and experiences, projecting future adoption use. This was a qualitative size of a good sample size. Use of the World Café form for data collection was most appropriate. Findings are meaning and thoughtful. The discussion of findings is informative. Improvements are possible to add a section on the theoretical foundations in the: value based social science and deliberative democracy to the introduction and framing the study on these key concepts rather than jump them in under methods and materials. Also, the introduction section would be better with questions for study towards the end and just before the methods section so readers can anticipate the answers under Findings.
We have moved some of the theoretical foundations from the methods section to the end of the introduction, in particular, the paragraph: "The World Café approach was developed in the 1990s by Juanita Brown and David Isaacs in collaboration with others [17]. This development paralleled the deliberative turn in democratic theory described by Dryzek [18] who argued that democratic legitimacy is supported by authentic deliberations on the part of those affected by a collective decision."
We have then added two sentences: “In using the World Café method, we brought together notions of informed publics, rich collaborative discussion in a hospitable space and collective knowledge made visible. In this setting, we asked a diverse group of older people to reflect on the actual and potential benefits and harms of using smart technologies in their lives as they age.”
We did not think it appropriate to move the questions for participants shown in Box 1 since these were a guide to discussions rather than an overarching research question.
Much more important, the Discussion section is way too long and quite repetitious some places. Authors should cut this down to by at least 2 pages. Some of that wring space “saved” should be used for a section on implications for research and practice (1/2 page) and another section on Limitations of the study and suggestions for further research (1/2 page). The conclusion needs revisiting to state the key findings in 5-7 sentences, with a statement of implication.
We believe that such a drastic pruning of the discussion would lose a great deal of the value in the paper and would mean that it would be a very different paper – more utilitarian and less reflective – than the one we have written. It is also not in the ambit of a “minor revision” which is what has been asked of us. We have made some deletions of text where we feel it does not disrupt the flow and richness of the discussion. However, we agree with the reviewer that some attention to the limitations and more detail in the conclusion about the findings and implications would be helpful.
Therefore, we have added a brief paragraph describing the limitations as follows:
“The study limitations include that this is an exploratory study focused on the views of older people and did not include the views of carers or families. Neither did it tease out completely the differences in attitudes due to age. More research is needed to understand the underpinning motivations for attitudes towards, and expectations of, smart assistive technologies for aging in place.”
Further, we have expanded the conclusion to state some of the key findings and implications of those findings as follows:
“Our findings suggest that developers and implementers of smart technologies should not underestimate the tech-savviness of older citizens but should work with users to facilitate ease of use and address concerns about privacy and confidentiality. Interestingly, the participants in our study did not see technology as an (external) labor saving tool, but as a (self) enabling tool: approaching technological solutions for common problems from that perspective may be important for the final acceptance and usefulness of technologies. The findings presented indicate that technologies for smart aging should be affordable, customized, and must not replace human contact and interaction. Similarly, although safety is important, the emphasis should be on supporting independent socially connected lives.”